# TensorNet: Cartesian Tensor Representations for Efficient Learning of Molecular Potentials

**Guillem Simeon**
Computational Science Laboratory
Universitat Pompeu Fabra
guillem.simeon@gmail.com

**Gianni De Fabritiis**
Computational Science Laboratory
Icrea, Universitat Pompeu Fabra, Acellera
g.defabritiis@gmail.com

## Abstract

The development of efficient machine learning models for molecular systems representation is becoming crucial in scientific research. We introduce Tensor-Net, an innovative $O(3)$-equivariant message-passing neural network architecture that leverages Cartesian tensor representations. By using Cartesian tensor atomic embeddings, feature mixing is simplified through matrix product operations. Furthermore, the cost-effective decomposition of these tensors into rotation group irreducible representations allows for the separate processing of scalars, vectors, and tensors when necessary. Compared to higher-rank spherical tensor models, TensorNet demonstrates state-of-the-art performance with significantly fewer parameters. For small molecule potential energies, this can be achieved even with a single interaction layer. As a result of all these properties, the model's computational cost is substantially decreased. Moreover, the accurate prediction of vector and tensor molecular quantities on top of potential energies and forces is possible. In summary, TensorNet's framework opens up a new space for the design of state-of-the-art equivariant models.

## 1 Introduction

Interatomic potential modeling using neural networks is an emerging research area that holds great promise for revolutionizing molecular simulation and drug discovery pipelines [1; 2; 3; 4]. The conventional trade-off between accuracy and computational cost can be bypassed by training models on highly precise data [5; 6; 7; 8]. Current state-of-the-art methodologies rely on equivariant graph neural networks (GNNs) [9; 10] and message-passing neural network (MPNNs) frameworks [11; 12], where internal atomic representations incorporate well-defined transformation properties characteristic of physical systems. The integration of equivariant features into neural network interatomic potentials has led to remarkable improvements in accuracy, particularly when using higher-rank irreducible representations of the orthogonal group $O(3)$—which encompasses reflections and rotations in 3D space—in the form of spherical tensors [13; 14]. Although lower-rank Cartesian representations (scalars and vectors) have been employed [15; 16], their success has been limited compared to state-of-the-art spherical models [17; 18; 19]. MPNNs typically necessitate a substantial number of message-passing iterations, and models based on irreducible representations are generally computationally demanding due to the need to compute tensor products, even though some successful alternative has been put forward [20].

The pursuit of computationally efficient approaches for incorporating higher-rank equivariance is essential. In this paper, we introduce a novel $O(3)$-equivariant architecture that advances the integration of Cartesian representations by utilizing Cartesian rank-2 tensors, represented as 3x3 matrices. We demonstrate that this method achieves state-of-the-art performance comparable to higher-rank spherical models while having a reduced computational cost. This efficiency is realized

37th Conference on Neural Information Processing Systems (NeurIPS 2023).

through the cheap decomposition of Cartesian rank-2 tensors into their irreducible components under the rotation group and the ability to mix features using straightforward 3x3 matrix products. Additionally, our model requires fewer message-passing steps and eliminates the need for explicit construction of many-body terms. The architecture also facilitates the direct prediction of tensor quantities, enabling the modeling of molecular phenomena where such quantities are relevant. In summary, we propose an alternative framework for developing efficient and accurate equivariant models.

## 2 Background and related work

**Equivariance.** A function $f$ between two vector spaces $X$ and $Y$, $f : X \to Y$, is said to be equivariant to the action of some abstract group $G$ if it fulfills

$$D_Y[g]f(x) = f(D_X[g]x), \tag{1}$$

for all elements $g \in G$ and $x \in X$, where $D_X[g]$ and $D_Y[g]$ denote the representations of $g$ in $X$ and $Y$, respectively. Equivariant neural networks used in neural network potentials focus on equivariance under the action of translations and the orthogonal group $O(3)$ in $\mathbb{R}^3$, the latter one being comprised by the rotation group $SO(3)$ and reflections, and regarded as a whole as the Euclidean group $E(3)$.

**Cartesian tensors and irreducible tensor decomposition.** Tensors are algebraic objects that generalize the notion of vectors. In the same way, as vectors change their components with respect to some basis under the action of a rotation $R \in SO(3)$, $\mathbf{v}' = R\mathbf{v}$, a rank-$k$ Cartesian tensor $T$ can be (very) informally regarded as a multidimensional array with $k$ indices, where each index transforms as a vector under the action of a rotation. In particular, a rank-2 tensor transformation under a rotation can be written in matrix notation as $T' = R\,TR^{\mathrm{T}}$, where $R^{\mathrm{T}}$ denotes the transpose of the rotation matrix $R$. In this paper, we will restrict ourselves to rank-2 tensors. Moreover, any rank-2 tensor $X$ defined on $\mathbb{R}^3$ can be rewritten in the following manner [21]

$$X = \frac{1}{3}\mathrm{Tr}(X)\mathrm{Id} + \frac{1}{2}(X - X^{\mathrm{T}}) + \frac{1}{2}(X + X^{\mathrm{T}} - \frac{2}{3}\mathrm{Tr}(X)\mathrm{Id}), \tag{2}$$

where $\mathrm{Tr}(X) = \sum_i X_{ii}$ is the trace operator and $\mathrm{Id}$ is the identity matrix. The first term is proportional to the identity matrix, the second term is a skew-symmetric contribution, and the last term is a symmetric traceless contribution. It can be shown that expression (2) is a decomposition into separate representations that are not mixed under the action of the rotation group [21]. In particular, the first component $I^X \equiv \frac{1}{3}\mathrm{Tr}(X)\mathrm{Id}$ has only 1 degree of freedom and is invariant under rotations, that is, it is a scalar; the second term $A^X \equiv \frac{1}{2}(X - X^{\mathrm{T}})$ has 3 independent components since it is a skew-symmetric tensor, which can be shown to rotate as a vector; and $S^X \equiv \frac{1}{2}(X + X^{\mathrm{T}} - \frac{2}{3}\mathrm{Tr}(X)\mathrm{Id})$ rotates like a rank-2 tensor and has 5 independent components, since a symmetric tensor has six independent components but the traceless condition removes one degree of freedom. In terms of representation theory, the 9-dimensional representation (a 3x3 matrix) has been reduced to irreducible representations of dimensions 1, 3, and 5 [21]. We will refer to $X$ as a full tensor and to the components $I^X, A^X, S^X$ as scalar, vector, and tensor features, respectively.

**Message passing neural network potentials.** Message-passing neural networks (MPNNs) have been successfully applied to the prediction of molecular potential energies and forces [11]. Atoms are represented by graph nodes, which are embedded in three-dimensional Euclidean space, and edges between nodes are built according to their relative proximity after the definition of some cutoff radius. The neural network uses atomic and geometric information, such as distances, angles or relative position vectors, to learn useful node representations by recursively propagating, aggregating, and transforming features from neighboring nodes [22; 23; 15]. In the case of neural network potentials, after several rounds of message passing and feature transformations, node features are mapped to single per-atom scalar quantities which are atomic contributions to the total energy of the molecule. These energy contributions depend in a very complex way on the states of other atoms, and therefore MPNNs can be regarded as some learnable approximation to the many-body potential energy function. However, these neural networks have typically needed a substantially large amount of message-passing steps (up to 6 in some cases) [16; 17].

**Equivariant models.** Initially, since the potential energy is a scalar quantity, atomic features were built using geometric information which is invariant to translations, reflections, and rotations, such as in SchNet [22], DimeNet [24; 25], PhysNet [26], SphereNet [27] and GemNet [23]. Nevertheless,

it has been shown that the inclusion of equivariant internal features leads to substantially better performances and data efficiency [14; 28]. In equivariant GNNs, internal features transform in a specified way under some group action. Molecules are physical systems embedded in three-dimensional Euclidean space, and their properties display well-defined behaviors under transformations such as translations, reflections, and rotations. Therefore, when predicting molecular properties, the group of interest is the orthogonal group in three dimensions $O(3)$, that is, rotations and reflections of the set of atoms in 3D space.

In models such as NewtonNet [29], EGNN [30], PaiNN [15], the Equivariant Transformer [16] and SAKE [31], Cartesian vector features are used on top of invariant features. These vector features are built using relative position vectors between input atoms, in such a way that when the input atomic Cartesian coordinates $\mathbf{r}$ are transformed under the action of some $R \in O(3)$ represented by a 3x3 matrix, $\mathbf{r} \to R\mathbf{r}$, internal vector features and vector outputs $\mathbf{v}$ transform accordingly, $\mathbf{v} \to R\mathbf{v}$. Other models such as Cormorant [13], Tensor Field Networks [32], NequIP [17], Allegro [18], BOTNet [19] and MACE [20], work directly with internal features that are irreducible representations of the group $O(3)$, which can be labeled by some $l \in \mathbb{N}$ (including $l = 0$), and with dimensions $2l + 1$. The representations $l = 0$ and $l = 1$ correspond to scalars and vectors, respectively. In this case, under a transformation $R \in O(3)$ of the input coordinates $\mathbf{r} \to R\mathbf{r}$, internal features $h_{lm}(\mathbf{r})$ transform as $h_{lm}(R\mathbf{r}) = \sum_{m'} D^l_{m'm}(R) \, h_{lm'}(\mathbf{r})$, where $D^l_{m'm}(R) \in \mathbb{R}^{(2l+1)\times(2l+1)}$ is an order $l$ Wigner D-matrix. In this case, features are rank-$l$ spherical tensors or pseudotensors, depending on their parity. The decomposition of a Cartesian tensor described in (2) and the irreducible representations in terms of spherical tensors are directly related by a change of basis [21]. To generate new features that satisfy $O(3)$-equivariance, these are built by means of tensor products involving Clebsch-Gordan coefficients and parity selection rules. In particular, models with features $l > 1$ such as NequIP, Allegro, BOTNet, and MACE have achieved state-of-the-art performances in benchmark datasets in comparison to all other MPNNs. However, the computation of tensor products in most of these models containing higher-rank tensors and pseudotensors can be expensive, especially when computing them in an edge-wise manner.

## 3 TensorNet's architecture

### 3.1 Operations respecting O(3)-equivariance

In this work, we propose the use of the 9-dimensional representation of rank-2 tensors (3x3 matrices). TensorNet operations are built to satisfy equivariance to the action of the orthogonal group $O(3)$: equivariance under $O(3)$ instead of the subgroup of rotations $SO(3)$ requires the consideration of the differences between tensors and pseudotensors. Tensors and pseudotensors are indistinguishable under rotations but display different behaviors under parity transformation, i.e. a reflection of the coordinate system through the origin. By definition, scalars, rank-2 tensors and in general all tensors of even rank do not flip their sign, and their parity is said to be even; on the contrary, vectors and tensors of odd rank have odd parity and flip their sign under the parity transformation. Pseudoscalars, pseudovectors, and pseudotensors have precisely the opposite behavior. Necessary derivations for the following subsections can be found in the Appendix (section A.2).

**Composition from irreducible representations.** The previously described irreducible decomposition of a tensor in Eq. 2 is with respect to the rotation group $SO(3)$. Building a tensor-like object that behaves appropriately under rotations can be achieved by composing any combination of scalars, vectors, tensors, and their parity counterparts. However, in neural network potential settings, the most direct way to produce tensors is by means of relative position *vectors* and, in general, it is preferred for the neural network to be able to predict vectors rather than pseudovectors. One has the possibility to initialize full tensor representations from the composition of scalars, vectors encoded in skew-symmetric matrices, and symmetric traceless tensors. For instance, if one considers some vector $\mathbf{v} = (v^x, v^y, v^z)$, one can build a well-behaved tensor $X$ under rotations by composing $X = I + A + S$,

$$I = f(||\mathbf{v}||)\mathrm{Id}, \quad A = \begin{pmatrix} 0 & v^z & -v^y \\ -v^z & 0 & v^x \\ v^y & -v^x & 0 \end{pmatrix}, \quad S = \mathbf{v}\mathbf{v}^{\mathrm{T}} - \frac{1}{3}\mathrm{Tr}(\mathbf{v}\mathbf{v}^{\mathrm{T}})\mathrm{Id}, \tag{3}$$

where $f$ is some function and $\mathbf{v}\mathbf{v}^{\mathrm{T}}$ denotes the outer product of the vector with itself. In this case, under parity the vector transforms as $\mathbf{v} \to -\mathbf{v}$, and it is explicit that $I$ and $S$ remain invariant, while $A \to -A = A^{\mathrm{T}}$, and the full tensor $X$ transforms as $X = I + A + S \to X' = I + A^{\mathrm{T}} + S = X^{\mathrm{T}}$, since $I$ and $S$ are symmetric matrices. Therefore, one concludes that when initializing the skew-symmetric part $A$ from vectors, not pseudovectors, parity transformation produces the transposition of full tensors.

**Invariant weights and linear combinations.** One can also modify some tensor $X = I + A + S$ by multiplying invariant quantities to the components, $X' = f_I I + f_A A + f_S S$, where $f_I, f_A$ and $f_S$ can be constants or invariant functions. This modification of the tensor does not break the tensor transformation rule under the action of rotations and preserves the parity of the individual components given that $f_I, f_A$ and $f_S$ are scalars (learnable functions of distances or vector norms, for example), not pseudoscalars. Also, from this property and the possibility of building full tensors from the composition of irreducible components, it follows that linear combinations of scalars, vectors, and tensors generate new full tensor representations that behave appropriately under rotations. Regarding parity, linear combinations preserve the original parity of the irreducible components given that all terms in the linear combination have the same parity. Therefore, given a set of irreducible components $I_j, A_j, S_j$ with $j \in \{0, 1, ..., n-1\}$, one can build full tensors $X'_i$

$$X'_i = \sum_{j=0}^{n-1} w^I_{ij} I_j + \sum_{j=0}^{n-1} w^A_{ij} A_j + \sum_{j=0}^{n-1} w^S_{ij} S_j, \tag{4}$$

where $w^I_{ij}, w^A_{ij}, w^S_{ij}$ can be learnable weights, in which case the transformation reduces to the application of three different linear layers without biases to inputs $I_j, A_j, S_j$.

**Matrix product.** Consider two tensors, $X$ and $Y$, and some rotation matrix $R \in \mathrm{SO}(3)$. Under the transformation $R$, the tensors become $RXR^{\mathrm{T}}$ and $RYR^{\mathrm{T}}$. The matrix product of these tensors gives a new object that also transforms like a tensor under the transformation, $XY \to RXR^{\mathrm{T}}RYR^{\mathrm{T}} = RXR^{-1}RYR^{\mathrm{T}} = R(XY)R^{\mathrm{T}}$, since for any rotation matrix $R$, $R^{\mathrm{T}} = R^{-1}$. Taking into account their irreducible decomposition $X = I^X + A^X + S^X$ and $Y = I^Y + A^Y + S^Y$, the matrix product $XY$ consists of several matrix products among rotationally independent sectors $(I^X + A^X + S^X)(I^Y + A^Y + S^Y)$. These products will contribute to the different parts of the irreducible decomposition $XY = I^{XY} + A^{XY} + S^{XY}$. Therefore, one can regard the matrix product as a way of combining scalar, vector, and tensor features to obtain new features. However, when assuming that the skew-symmetric parts are initialized from vectors, this matrix product mixes components with different parities, and resulting components $I^{XY}, A^{XY}, S^{XY}$ would not have a well-defined behavior under parity (see Appendix, section A.2). To achieve O(3)-equivariance, we propose the use of the matrix products $XY + YX$. Under parity $X \to X^{\mathrm{T}}, Y \to Y^{\mathrm{T}}$, and one can show that

$$I^{X^{\mathrm{T}}Y^{\mathrm{T}}+Y^{\mathrm{T}}X^{\mathrm{T}}} = I^{XY+YX}, \quad A^{X^{\mathrm{T}}Y^{\mathrm{T}}+Y^{\mathrm{T}}X^{\mathrm{T}}} = -A^{XY+YX}, \quad S^{X^{\mathrm{T}}Y^{\mathrm{T}}+Y^{\mathrm{T}}X^{\mathrm{T}}} = S^{XY+YX}, \tag{5}$$

that is, the scalar and symmetric traceless parts have even parity, and the skew-symmetric part has odd parity. The irreducible decomposition of the expression $XY + YX$ preserves the rotational and parity properties of the original components and, therefore, it is an O(3)-equivariant operation. We finally note that one can produce O(3)-invariant quantities from full tensor representations or their components by taking their Frobenius norm $\mathrm{Tr}(X^{\mathrm{T}}X) = \mathrm{Tr}(XX^{\mathrm{T}}) = \sum_{ij} |X_{ij}|^2$.

## 3.2 Model architecture

In this work we propose a model that learns a set of Cartesian full tensor representations $X^{(i)}$ (3x3 matrices) for every atom $(i)$, from which atomic or molecular properties can be predicted, using as inputs atomic numbers $z_i$ and atomic positions $\mathbf{r}_i$. We mainly focus on the prediction potential energies and forces, even though we provide in Section 4 experiments demonstrating the ability of TensorNet to accurately predict up to rank-2 physical quantities. Atomic representations $X^{(i)}$ can be decomposed at any point into scalar, vector and tensor contributions $I^{(i)}, A^{(i)}, S^{(i)}$ via (2), and TensorNet can be regarded as operating with a physical inductive bias akin to the usual decomposition of interaction energies in terms of monopole, dipole and quadrupole moments [13]. We refer the reader to Figure 1 and the Appendix (section A.1) for diagrams of the methods and the architecture.

**Embedding.** By defining a cutoff radius $r_c$, we obtain vectors $\mathbf{r}_{ij} = \mathbf{r}_j - \mathbf{r}_i$ between central atom $i$ and neighbors $j$ within a distance $r_c$. We initialize per-edge scalar features using the identity

matrix $I_0^{(ij)} = \mathrm{Id}$, and per-edge vector and tensor features using the normalized edge vectors $\hat{r}_{ij} = \mathbf{r}_{ij}/\|\mathbf{r}_{ij}\| = (\hat{r}_{ij}^x, \hat{r}_{ij}^y, \hat{r}_{ij}^z)$. We create a symmetric traceless tensor from the outer product of $\hat{r}_{ij}$ with itself, $S_0^{(ij)} \equiv \hat{r}_{ij}\hat{r}_{ij}^{\mathrm{T}} - \frac{1}{3}\mathrm{Tr}(\hat{r}_{ij}\hat{r}_{ij}^{\mathrm{T}})\mathrm{Id}$, and vector features are initialized by identifying the independent components of the skew-symmetric contribution with the components of $\hat{r}_{ij}$ as denoted in (3), getting for every edge $(ij)$ initial irreducible components $I_0^{(ij)}, A_0^{(ij)}, S_0^{(ij)}$. To

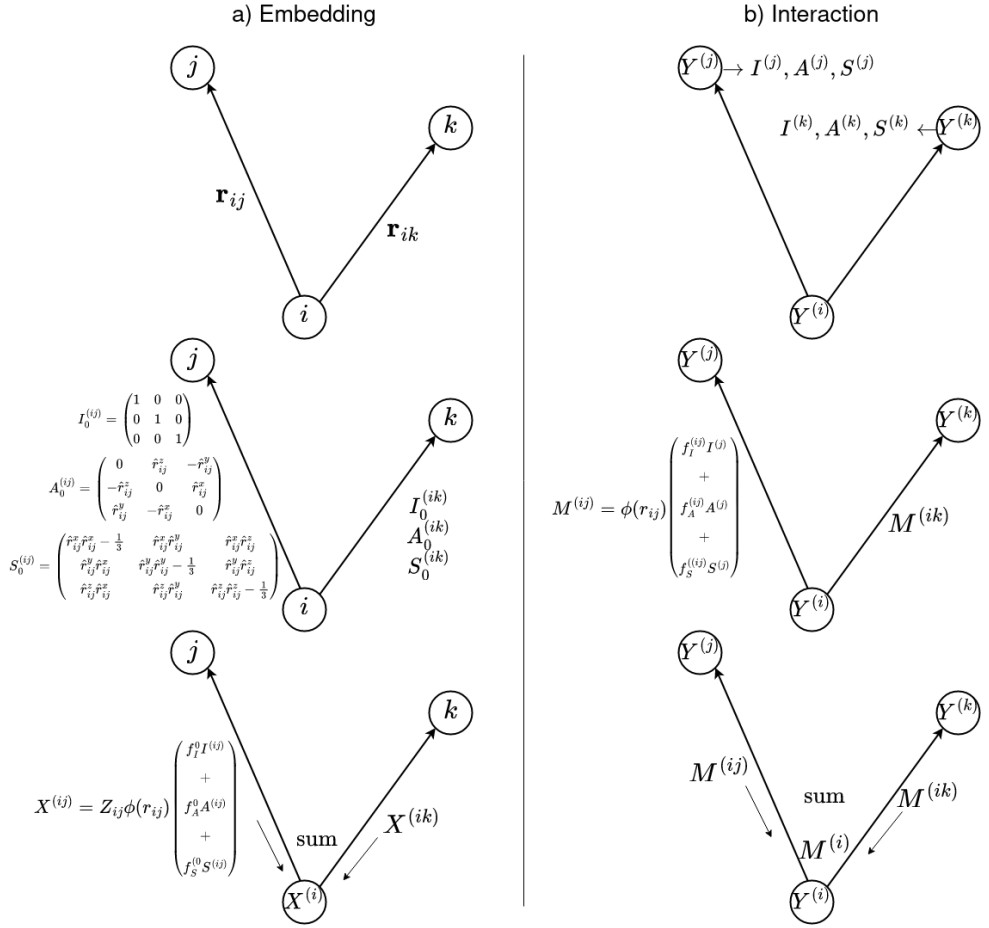

Figure 1: Key steps, from top to bottom, in the embedding and interaction modules for some central atom $i$ and neighbors $j$ and $k$ found within the cutoff radius. a) Relative position vectors are used to initialize edge-wise tensor components, modified using edge-wise invariant functions, and summed to obtain node-wise full tensors. b) Node full tensors are decomposed and weighted with edge invariant functions to obtain pair-wise messages, and summed to obtain node-wise aggregated messages, which will interact with receiving node's full tensors via matrix product.

encode interatomic distance and atomic number information in the tensor representations we use an embedding layer that maps the atomic number of every atom $z_i$ to $n$ invariant features $Z_i$, and expand interatomic distances $r_{ij}$ to $d$ invariant features by means of an expansion in terms of exponential radial basis functions

$$e_k^{\mathrm{RBF}}(r_{ij}) = \exp\left(-\beta_k(\exp(-r_{ij}) - \mu_k)^2\right), \qquad (6)$$

where $\beta_k$ and $\mu_k$ are fixed parameters specifying the center and width of radial basis function $k$. The $\mu$ vector is initialized with values equally spaced between $\exp(-r_c)$ and 1, and $\beta$ is initialized as $\left(2d^{-1}(1 - \exp\left(-r_c\right))\right)^{-2}$ for all $k$ as proposed in [26]. After creating $n$ identical copies of initial components $I_0^{(ij)}, A_0^{(ij)}, S_0^{(ij)}$ ($n$ feature channels), for every edge $(ij)$ we map with a linear layer the concatenation of $Z_i$ and $Z_j$ to $n$ pair-wise invariant representations $Z_{ij}$, and the radial basis functions are further expanded to $n$ scalar features by using three different linear layers to obtain

$$f_I^0 = W^I(e^{\mathrm{RBF}}(r_{ij})) + b^I, \quad f_A^0 = W^A(e^{\mathrm{RBF}}(r_{ij})) + b^A, \quad f_S^0 = W^S(e^{\mathrm{RBF}}(r_{ij})) + b^S, \quad (7)$$

$$X^{(ij)} = \phi(r_{ij})Z_{ij}\left(f_I^0 I_0^{(ij)} + f_A^0 A_0^{(ij)} + f_S^0 S_0^{(ij)}\right), \tag{8}$$

where the cutoff function is given by $\phi(r_{ij}) = \frac{1}{2}\left(\cos\left(\frac{\pi r_{ij}}{r_c}\right) + 1\right)$ when $r_{ij} \leq r_c$ and 0 otherwise. That is, $n$ edge-wise tensor representations $X^{(ij)}$ are obtained, where the channel dimension has not been written explicitly. Then, we get atom-wise tensor representations by aggregating all neighboring edge-wise features. At this point, the invariant norms $||X|| \equiv \mathrm{Tr}(X^{\mathrm{T}}X)$ of atomic representations $X^{(i)}$ are computed and fed to a normalization layer, a multilayer perceptron, and a SiLU activation to obtain three different O(3)-invariant functions per channel,

$$f_I^{(i)}, f_A^{(i)}, f_S^{(i)} = \mathrm{SiLU}(\mathrm{MLP}(\mathrm{LayerNorm}(||X^{(i)}||))), \tag{9}$$

which, after the decomposition of tensor embeddings into their irreducible representations, are then used to modify component-wise linear combinations to obtain the final atomic tensor embeddings

$$X^{(i)} \leftarrow f_I^{(i)}W^I I^{(i)} + f_A^{(i)}W^A A^{(i)} + f_S^{(i)}W^S S^{(i)}. \tag{10}$$

**Interaction and node update.** We start by normalizing each node's tensor representation $X^{(i)} \leftarrow X^{(i)}/(||X^{(i)}|| + 1)$ and decomposing this representation into scalar, vector, and tensor features. We next transform these features $I^{(i)}, A^{(i)}, S^{(i)}$ by computing independent linear combinations, $Y^{(i)} = W^I I^{(i)} + W^A A^{(i)} + W^S S^{(i)}$. In parallel, edge distances' radial basis expansions are fed to a multilayer perceptron and a SiLU activation to transform them into tuples of three invariant functions per channel weighted with the cutoff function $\phi(r_{ij})$,

$$f_I^{(ij)}, f_A^{(ij)}, f_S^{(ij)} = \phi(r_{ij})\mathrm{SiLU}(\mathrm{MLP}(e^{\mathrm{RBF}}(r_{ij}))). \tag{11}$$

At this point, after decomposition of node features $Y^{(i)}$, we define the messages sent from neighbors $j$ to central atom $i$ as $M^{(ij)} = f_I^{(ij)}I^{(j)} + f_A^{(ij)}A^{(j)} + f_S^{(ij)}S^{(j)}$, which get aggregated into $M^{(i)} = \sum_{j \in \mathcal{N}(i)} M^{(ij)}$. We use the irreducible decomposition of matrix products $Y^{(i)}M^{(i)} + M^{(i)}Y^{(i)}$ between node embeddings and aggregated messages to generate new atomic scalar, vector, and tensor features. New features are generated in this way to guarantee the preservation of the original parity of scalar, vector, and tensor features. These new representations $I^{(i)}, A^{(i)}, S^{(i)}$ are individually normalized dividing by $||I^{(i)} + A^{(i)} + S^{(i)}|| + 1$ and are further used to compute independent linear combinations to get $Y^{(i)} \leftarrow W^I I^{(i)} + W^A A^{(i)} + W^S S^{(i)}$. A residual update $\Delta X^{(i)}$ for original embeddings $X^{(i)}$ is computed with the parity-preserving matrix polynomial $\Delta X^{(i)} = Y^{(i)} + (Y^{(i)})^2$, to eventually obtain updated representations $X^{(i)} \leftarrow X^{(i)} + \Delta X^{(i)}$.

**Scalar output.** The Frobenius norm $\mathrm{Tr}(X^{\mathrm{T}}X)$ of full tensor representations and components in TensorNet is O(3)-invariant. For molecular potential predictions, total energy $U$ is computed from atomic contributions $U^{(i)}$ which are simply obtained by using the concatenated final norms of every atom's scalar, vector, and tensor features $||I^{(i)}||, ||A^{(i)}||, ||S^{(i)}||$,

$$U^{(i)} = \mathrm{MLP}(\mathrm{LayerNorm}([\,||I^{(i)}||, ||A^{(i)}||, ||S^{(i)}||\,])), \tag{12}$$

obtaining forces via backpropagation.

**Vector output.** Since interaction and update operations preserve the parity of tensor components, the skew-symmetric part of any full tensor representation $X$ in TensorNet is guaranteed to be a vector, not a pseudovector. Therefore, from the antisymmetrization $A^X$, one can extract vectors $\mathbf{v} = (v_x, v_y, v_z)$ by means of the identification given in (3).

**Tensor output.** Taking into account that rank-2 tensors have even parity and the skew-symmetric part $A$ in TensorNet is a vector, not a pseudovector, one might need to produce pseudovector features before rank-2 tensor predictions can be built by combining irreducible representations. This can be easily done by obtaining two new vector features with linear layers, $A^{(1)} = W^{(1)}A$ and $A^{(2)} = W^{(2)}A$, and computing $\frac{1}{2}(A^{(1)}A^{(2)} - (A^{(1)}A^{(2)})^{\mathrm{T}})$, which is skew-symmetric, rotates like a vector, and is invariant under parity, the simultaneous transposition of $A^{(1)}$ and $A^{(2)}$.

## 4    Experiments and results

We refer the reader to the Appendix (section A.3) for further training, data set and experimental details.

**QM9: Chemical diversity.** To assess TensorNet's accuracy in the prediction of energy-related molecular properties with a training set of varying chemical composition we used QM9 [33]. We trained TensorNet to predict: $U_0$, the internal energy of the molecule at 0 K; $U$, the internal energy at 298.15 K; $H$, the enthalpy, also at 298.15 K; and $G$, the free energy at 298.15 K. Results can be found in Table 1, which show that TensorNet outperforms Allegro [18], and MACE [34] on $U_0$, $U$ and $H$. Remarkably, this is achieved with 23% of Allegro's parameter count. Furthermore, TensorNet uses only scalar, vector and rank-2 tensor features, as opposed to Allegro, which uses also their parity counterparts, and without the need of explicitly taking into account many-body terms as done in MACE.

Table 1: **QM9 results.** Mean absolute error on energy-related molecular properties from the QM9 dataset, in meV, averaged over different splits. Parameter counts for some models are found between parentheses.

| **Property** | DimeNet++ [25] | ET [16] | PaiNN [15] | Allegro (17.9M)[18] | MACE [34] | TensorNet (4.0M) |
|---|---|---|---|---|---|---|
| $U_0$ | 6.3 | 6.2 | 5.9 | 4.7 | 4.1 | **3.9(1)** |
| $U$ | 6.3 | 6.3 | 5.7 | 4.4 | 4.1 | **3.9(1)** |
| $H$ | 6.5 | 6.5 | 6.0 | 4.4 | 4.7 | **4.0(1)** |
| $G$ | 7.6 | 7.6 | 7.4 | 5.7 | **5.5** | 5.7(1) |

**rMD17: Conformational diversity.** We also benchmarked TensorNet on rMD17 [35], the revised version of MD17 [36; 37], a data set of small organic molecules in which energies and forces were obtained by running molecular dynamics simulations with DFT. We report the results in Table 2. In the case of energies, TensorNet with two interaction layers (2L) is the model that achieves state-of-the-art accuracy for the largest number of molecules (6 out of 10), outperforming all other spherical models for benzene, with a parameter count of 770k. Energy errors are also within the range of other spherical models, except for ethanol and aspirin, and reach state-of-the-art accuracy for the case of toluene, with just one interaction layer (1L) and a parameter count of 535k. Force errors for 2L are also mostly found within the ranges defined by other spherical models, except for ethanol, aspirin, and salicylic acid, in which case these are slightly higher. However, for one interaction layer, force errors are increased and in most cases found outside of the range of accuracy of the other spherical models. We note that the smallest spherical models have approximately 2.8M parameters, and therefore TensorNet results are achieved with reductions of 80% and 70% in the number of parameters for 1L and 2L, respectively. Also, TensorNet is entirely based at most on rank-2 tensors.

**SPICE, ANI1x, COMP6: Compositional and conformational diversity.** To obtain general-purpose neural network interatomic potentials, models need to learn simultaneously compositional and conformational degrees of freedom. In this case, data sets must contain a wide range of molecular systems as well as several conformations per system. To evaluate TensorNet's out-of-the-box performance without hyperparameter fine-tuning, we trained the light model with two interaction layers used on rMD17 on the SPICE [38] and ANI1x [39; 40] data sets using the proposed Equivariant Transformer's SPICE hyperparameters [41] (for ANI1x, in contrast to the SPICE model, we used 32 radial basis functions instead of 64, and a cutoff of 4.5Å instead of 10Å), and further evaluated ANI1x-trained models on the COMP6 benchmarks [39].

For SPICE, with a maximum force filter of 50.94 eV/Å $\approx$ 1 Ha/Bohr, TensorNet's mean absolute error in energies and forces are 25.0 meV and 40.7 meV/Å, respectively, while the Equivariant Transformer achieves 31.2 meV and 49.3 meV/Å. In this case, both models used a cutoff of 10Å. Results for ANI1x and model evaluations on COMP6 are found in Table 3. We note that for ANI1x training, which contains molecules with up to 63 atoms, TensorNet used a cutoff of 4.5Å. The largest rMD17 molecule is aspirin with 21 atoms. The light TensorNet model shows better generalization capabilities across all COMP6 benchmarks.

**Scalar, vector and tensor molecular properties for ethanol in a vacuum.** We next tested TensorNet performance for the simultaneous prediction of scalar, vector, and tensor molecular properties: potential energy, atomic forces, molecular dipole moments $\mu$, molecular polarizability tensors $\alpha$, and nuclear-shielding tensors $\sigma$, for the ethanol molecule in vacuum [15; 42]. We trained TensorNet to generate atomic tensor representations that can be used by different output modules to predict the

Table 2: **rMD17 results.** Energy (E) and forces (F) mean absolute errors in meV and meV/Å, averaged over different splits.

| Molecule | | TensorNet 1L (535k) | TensorNet 2L (770k) | NequIP [17] | Allegro [18] | BOTNet [19] | MACE [20] |
|---|---|---|---|---|---|---|---|
| Aspirin | E | 2.7 | 2.4 | 2.3 | 2.3 | 2.3 | **2.2** |
| | F | 10.2(2) | 8.9(1) | 8.2 | 7.3 | 8.5 | **6.6** |
| Azobenzene | E | 0.9 | **0.7** | **0.7** | 1.2 | **0.7** | 1.2 |
| | F | 3.8 | 3.1 | 2.9 | **2.6** | 3.3 | 3.0 |
| Benzene | E | 0.03 | **0.02** | 0.04 | 0.3 | 0.03 | 0.4 |
| | F | 0.3 | 0.3 | 0.3 | **0.2** | 0.3 | 0.3 |
| Ethanol | E | 0.5 | 0.5 | **0.4** | **0.4** | **0.4** | **0.4** |
| | F | 3.9(1) | 3.5 | 2.8 | **2.1** | 3.2 | **2.1** |
| Malonaldehyde | E | 0.8 | 0.8 | 0.8 | **0.6** | 0.8 | 0.8 |
| | F | 5.8(1) | 5.4 | 5.1 | **3.6** | 5.8 | 4.1 |
| Naphthalene | E | 0.3 | **0.2** | 0.9 | **0.2** | **0.2** | 0.5 |
| | F | 1.9 | 1.6 | 1.3 | **0.9** | 1.8 | 1.6 |
| Paracetamol | E | 1.5 | **1.3** | 1.4 | 1.5 | **1.3** | **1.3** |
| | F | 6.9 | 5.9(1) | 5.9 | 4.9 | 5.8 | **4.8** |
| Salicylic acid | E | 0.9 | 0.8 | **0.7** | 0.9 | 0.8 | 0.9 |
| | F | 5.4(1) | 4.6(1) | 4.0 | **2.9** | 4.3 | 3.1 |
| Toluene | E | **0.3** | **0.3** | **0.3** | 0.4 | 0.4 | 0.5 |
| | F | 2.0 | 1.7 | 1.6 | 1.8 | 1.9 | **1.5** |
| Uracil | E | 0.5 | **0.4** | **0.4** | 0.6 | **0.4** | 0.5 |
| | F | 3.6(1) | 3.1 | 3.1 | **1.8** | 3.2 | 2.1 |

Table 3: **ANI1x and COMP6 results.** Energy (E) and forces (F) mean absolute errors in meV and meV/Å for Equivariant Transformer and TensorNet models trained on ANI1x and evaluated on the ANI1x test set and the COMP6 benchmarks, averaged over different training splits. 43 meV = 1 kcal/mol.

| Model | | ANI1x | ANI-MD | GDB7-9 | GDB10-13 | DrugBank | Tripeptides | S66x8 |
|---|---|---|---|---|---|---|---|---|
| ET | E | 21.2 | 249.7 | 17.8 | 51.0 | 95.5 | 57.9 | 30.7 |
| | F | 42.0 | 50.8 | 29.2 | 57.4 | 47.7 | 37.9 | 19.0 |
| TensorNet | E | 17.3 | 69.9 | 14.3 | 36.0 | 42.4 | 40.0 | 27.1 |
| | F | 34.3 | 35.5 | 23.1 | 41.9 | 32.6 | 26.9 | 14.3 |

desired properties. The specific architecture of these output modules can be found in the Appendix (section A.3).

Table 4: **Ethanol in vacuum results.** Mean absolute error for the prediction of energies (E), forces (F), dipole moments ($\mu$), polarizabilities ($\alpha$), and chemical shifts for all elements ($\sigma_{all}$), averaged over different splits, with corresponding units between parentheses.

| Model | E (kcal/mol) | F (kcal/mol/Å) | $\mu$ (D) | $\alpha$ (Bohr$^3$) | $\sigma_{all}$ (ppm) |
|---|---|---|---|---|---|
| PaiNN [15] | 0.027 | 0.150 | **0.003** | 0.009 | - |
| FieldSchNet [42] | 0.017 | 0.128 | 0.004 | 0.008 | 0.169 |
| TensorNet | **0.008(1)** | **0.058(3)** | **0.003(0)** | **0.007(0)** | **0.139(4)** |

Results from Table 4 show that TensorNet can learn expressive atomic tensor embeddings from which multiple molecular properties can be simultaneously predicted. In particular, TensorNet's energy and force errors are approximately a factor of two and three smaller when compared to FieldSchNet [42] and PaiNN [15], respectively, while increasing the prediction accuracy for the other target molecular properties, with the exception of the dipole moment.

**Equivariance, interaction and cutoff ablations.** TensorNet can be straightforwardly modified such that features are SO(3)-equivariant and scalar predictions are SE(3)-invariant by modifying the matrix products in the interaction mechanism. An interaction product between node features and aggregated messages $2Y^{(i)}M^{(i)}$, instead of $Y^{(i)}M^{(i)} + M^{(i)}Y^{(i)}$, gives vector and tensor representations which are combinations of even and odd parity contributions. We refer the reader to the Supplementary Material for detailed derivations. Furthermore, the norms $\mathrm{Tr}(X^\mathrm{T}X)$ used to produce scalars will only be invariant under rotations, not reflections. This flexibility in the model allows us to study the changes in prediction accuracy when considering O(3) or SO(3) equivariant models. We also evaluated the impact on accuracy for two rMD17 molecules, toluene and aspirin, when modifying the receptive field of the model by changing the cutoff radius and the number of interaction layers, including the case of using the embedding and output modules alone, without interaction layers (0L), with results in Table 5.

Table 5: **Equivariance, interaction and cutoff ablations results.** Energy (E) and force (F) mean absolute errors in meV and meV/Å for rMD17 toluene and aspirin, averaged over different splits, varying the number of interaction layers, the cutoff radius, and the equivariance group.

| Molecule | | TensorNet 0L O(3) 4.5Å | 9Å | TensorNet 1L O(3) 4.5Å | 9Å | TensorNet 2L O(3) 4.5Å | 9Å | TensorNet 1L SO(3) 4.5Å | TensorNet 2L SO(3) 4.5Å |
|---|---|---|---|---|---|---|---|---|---|
| Toluene | E | 3.3 | 2.0 | 0.33 | 0.36 | 0.26 | 0.32 | 0.50 | 0.42 |
| | F | 15.7 | 11.5 | 2.0 | 2.2 | 1.7 | 2.0 | 2.9 | 2.4 |
| Aspirin | E | 9.8 | 7.8 | 2.7 | 2.9 | 2.4 | 2.8 | 3.7 | 3.4 |
| | F | 32.7 | 28.3 | 10.1 | 11.0 | 8.9 | 10.5 | 13.1 | 11.8 |

The inclusion or exclusion of equivariance and energy invariance under reflections has a significant impact on accuracy. The consideration of the full orthogonal group O(3), and therefore the physical symmetries of the true energy function, leads to higher accuracy for both energy and forces. Furthermore, the use of interaction products produces a drastic decrease in errors (note that TensorNet 1L 4.5Å and TensorNet 0L 9Å have the same receptive field). In line with rMD17 results, a second interaction layer in the case of $r_c = 4.5$Å gives an additional but more limited improvement in both energy and force errors. For forces, the use of a second interaction layer with $r_c = 9$Å encompassing the whole molecule provides a smaller improvement when compared to $r_c = 4.5$Å. We note that for 0L, when the model can be regarded as just a learnable aggregation of local atomic neighborhoods, TensorNet with both cutoff radii achieves for aspirin (the rMD17 molecule on which the model performs the worst) lower mean absolute errors than ANI (16.6 meV and 40.6 meV/Å) [7; 43] and SchNet (13.5 meV and 33.2 meV/Å) [22; 44].

**Computational cost.** We found that TensorNet exhibits high computational efficiency, even higher than an equivariant model using Cartesian vectors such as the Equivariant Transformer [16] in some cases. We provide inference times for single molecules with varying numbers of atoms in Table 6, and in Table 7 we show training steps per second when training on the ANI1x data set, containing molecules with up to 63 atoms.

For molecules containing up to ∼200 atoms, TensorNet 1L and 2L are faster or similar when compared to the ET, even when its number of message passing layers is reduced (ET optimal performance for MD17 was achieved with 6 layers, found to be one of the fastest neural network potentials in the literature [16; 45]), meaning that energy and forces on these molecules can be predicted with rMD17 state-of-the-art TensorNet models with a lower or similar computational cost than the reduced ET. For larger molecules with thousands of atoms, TensorNet 2L becomes significantly slower. However, TensorNet 1L, which still exhibits remarkable performance on rMD17 (see Table 1), performs on par the reduced ET in terms of speed even for Factor IX, containing 5807 atoms. For training on ANI1x, TensorNet 1L and 2L are faster or comparable to the ET up to a batch size of 64, being the speed for the 2L model being significantly slower for a batch size of 128. Nevertheless, the model with 1 interaction layer is still comparable to the reduced ET.

TensorNet's efficiency is given by properties that are in contrast to state-of-the-art equivariant spherical models. In particular, the use of Cartesian representations allows one to manipulate full tensors or their decomposition into scalars, vectors, and tensors at one's convenience, and Clebsch-

Table 6: **Inference time.** Inference time for energy and forces for single molecules (batch size of 1), in ms, on an NVIDIA GeForce RTX 4090.

| Molecule | $N$ | TensorNet 0L | TensorNet 1L | TensorNet 2L | ET 4L | ET 5L |
|---|---|---|---|---|---|---|
| Alanine dipeptide | 22 | 10.0 | 22.1 | 26.5 | 27.2 | 29.0 |
| Chignolin | 166 | 10.4 | 22.5 | 26.9 | 26.8 | 28.9 |
| DHFR | 2489 | 27.3 | 66.9 | 106.7 | 52.4 | 67.7 |
| Factor IX | 5807 | 53.1 | 149.8 | 248.6 | 110.6 | 136.4 |

Table 7: **Training speed.** Number of batch training steps per second for ANI1x dataset on an NVIDIA GeForce RTX 4090.

| Batch size | TensorNet 0L | TensorNet 1L | TensorNet 2L | ET 4L | ET 5L |
|---|---|---|---|---|---|
| 32 | 20.5 | 13.2 | 10.1 | 9.5 | 8.4 |
| 64 | 19.1 | 12.9 | 9.1 | 9.4 | 8.3 |
| 128 | 17.3 | 8.9 | 5.9 | 9.1 | 8.0 |

Gordan tensor products are substituted for simple 3x3 matrix products. As detailed in the model's architecture, state-of-the-art performance can be achieved by computing these matrix products after message aggregation (that is, at the node level) and using full tensor representations, without having to individually compute products between different irreducible components. When considering an average number of neighbors per atom $M$ controlled by the cutoff radius and the density, given that matrix products are performed after aggregation over neighbors, these do not scale with $M$. This is in contrast to spherical models, where tensor products are computed on edges, and therefore display a worse scaling with the number of neighbors $M$, that is, a worse scaling when increasing the cutoff radius at fixed density. Also, the use of higher-order many-body messages or many message-passing steps is not needed.

## 5 Conclusions and limitations

We have presented TensorNet, a novel $O(3)$-equivariant message-passing architecture leveraging Cartesian tensors and their irreducible representations. We showed that even though the model is limited to the use of rank-2 tensors, in contrast to other spherical models, it achieves state-of-the-art performance on QM9 and rMD17 with a reduced number of parameters, few message-passing steps, and it exhibits a low computational cost. Furthermore, the model is able to accurately predict vector and rank-2 tensor molecular properties on top of potential energies and forces. Nevertheless, the prediction of higher-rank quantities is directly limited by our framework. However, given the benefits of the formalism for the construction of a machine learning potential, TensorNet can be used as an alternative for the exploration of the design space of efficient equivariant models. TensorNet can be found in `https://github.com/torchmd/torchmd-net`.

## Acknowledgments and Disclosure of Funding

We thank Raul P. Pelaez for fruitful discussions. GS is financially supported by Generalitat de Catalunya's Agency for Management of University and Research Grants (AGAUR) PhD grant FI-2-00587. This project has received funding from the European Union's Horizon 2020 research and innovation programme under grant agreement No. 823712 (RG, AV, RF); the project PID2020-116564GB-I00 has been funded by MCIN / AEI / 10.13039/501100011033. Research reported in this publication was supported by the National Institute of General Medical Sciences (NIGMS) of the National Institutes of Health under award number GM140090. The content is solely the responsibility of the authors and does not necessarily represent the official views of the National Institutes of Health.

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

# A Appendix

## A.1 Architecture diagrams

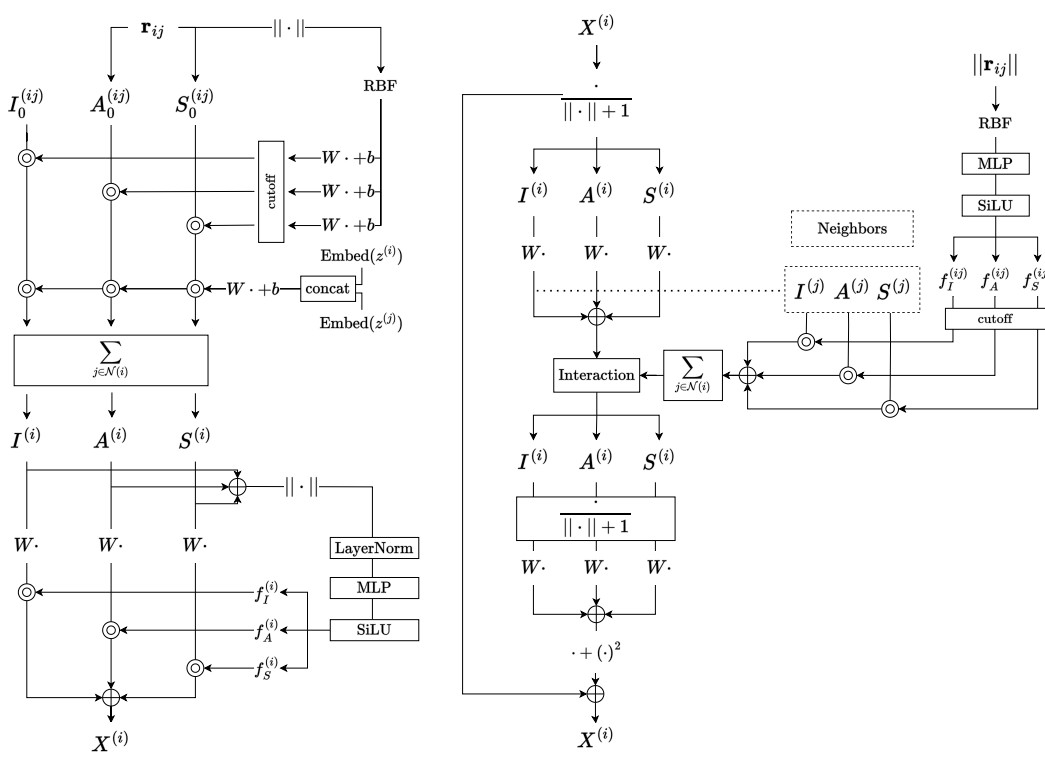

(a) Embedding                    (b) Interaction and node update

Figure A.1: Diagrams of the embedding and interaction and node update modules in TensorNet, descibed in Section 3.2. Models are built from an initial embedding module, the concatenation of several interaction layers, and the output module. The dotted line in Interaction and node update indicates the point where, after node level transformations, neighboring atom's features are used to create messages.

## A.2 Proofs and derivations

### Parity and interaction products

Consider some tensor $X$, with its corresponding decomposition $I^X + A^X + S^X$ and initialized as in TensorNet (see Section 3.2), that is, with vector features obtained directly by the identification of vector components with matrix entries (3). When considering the parity transformation, the components of the vector used to build the initial features flip sign, and for any full tensor initialized in this way

$$X = I^X + A^X + S^X \rightarrow X' = I^X - A^X + S^X = X^{\mathrm{T}}. \tag{A.1}$$

Consider now another tensor $Y = I^Y + A^Y + S^Y$ initialized in the same way, and the product $XY = (I^X + A^X + S^X)(I^Y + A^Y + S^Y)$. One can compute the resulting decomposition $I^{XY} = \frac{1}{3}\mathrm{Tr}(XY)\mathrm{Id}$, $A^{XY} = \frac{1}{2}(XY - (XY)^{\mathrm{T}}) = \frac{1}{2}(XY - Y^{\mathrm{T}}X^{\mathrm{T}})$ and $S^{XY} = \frac{1}{2}(XY + (XY)^{\mathrm{T}} - \frac{2}{3}\mathrm{Tr}(XY)\mathrm{Id}) = \frac{1}{2}(XY + Y^{\mathrm{T}}X^{\mathrm{T}} - \frac{2}{3}\mathrm{Tr}(XY)\mathrm{Id})$ from the individual decompositions of $X$ and $Y$. Taking into account that transposition is equivalent to sign reversal of skew-symmetric components, one obtains after some matrix algebra

$$I^{XY} = \frac{1}{3}\Bigg(\underbrace{\text{Tr}(I^X I^Y + A^X A^Y + S^X S^Y)}_{\text{scalar}} + \underbrace{\text{Tr}(A^X S^Y + S^X A^Y)}_{=0} +$$

$$+ \underbrace{\text{Tr}(I^X S^Y + S^X I^Y + I^X A^Y + A^X I^Y)}_{=0}\Bigg)\text{Id}, \tag{A.2}$$

$$A^{XY} = \underbrace{I^X A^Y + A^X I^Y + \frac{1}{2}(A^X S^Y - (A^X S^Y)^{\text{T}}) + \frac{1}{2}(A^Y S^X - (A^Y S^X)^{\text{T}})}_{\text{vector}} +$$

$$+ \underbrace{\frac{1}{2}(A^X A^Y - (A^X A^Y)^{\text{T}}) + \frac{1}{2}(S^X S^Y - (S^X S^Y)^{\text{T}})}_{\text{pseudovector}}, \tag{A.3}$$

$$S^{XY} = \underbrace{I^X I^Y - \frac{1}{3}\text{Tr}(I^X I^Y)\text{Id}}_{=0} +$$

$$+ \underbrace{\frac{1}{2}(A^X S^Y + (A^X S^Y)^{\text{T}} - \frac{2}{3}\text{Tr}(A^X S^Y)\text{Id}) + \frac{1}{2}(S^X A^Y + (S^X A^Y)^{\text{T}} - \frac{2}{3}\text{Tr}(S^X A^Y)\text{Id})}_{\text{pseudotensor}}$$

$$+ \underbrace{I^X S^Y + S^X I^Y + \frac{1}{2}(A^X A^Y + (A^X A^Y)^{\text{T}} - \frac{2}{3}\text{Tr}(A^X A^Y)\text{Id}) + \frac{1}{2}(S^X S^Y + (S^X S^Y)^{\text{T}} - \frac{2}{3}\text{Tr}(S^X S^Y)\text{Id})}_{\text{tensor}},$$

$$\tag{A.4}$$

where we have made explicit that the product $XY$ gives rise to both even parity (transposing $A^X$ and $A^Y$ does not flip the sign of the expression) and odd parity (transposing $A^X$ and $A^Y$ does flip the sign of the expression) skew-symmetric and symmetric traceless contributions. However, when considering $XY + YX$, one gets

$$I^{XY+YX} = \frac{1}{3}\Bigg(\underbrace{\text{Tr}(2I^X I^Y + A^X A^Y + (A^X A^Y)^{\text{T}} + S^X S^Y + (S^X S^Y)^{\text{T}})}_{\text{scalar}}\Bigg)\text{Id} \tag{A.5}$$

$$A^{XY+YX} = \underbrace{2I^X A^Y + 2A^X I^Y + (A^X S^Y - (A^X S^Y)^{\text{T}}) + (A^Y S^X - (A^Y S^X)^{\text{T}})}_{\text{vector}} +$$

$$+ \underbrace{\frac{1}{2}(A^X A^Y - A^Y A^X + A^Y A^X - A^X A^Y)}_{=0} +$$

$$+ \underbrace{\frac{1}{2}(S^X S^Y - S^Y S^X + S^Y S^X - S^X S^Y)}_{=0}, \tag{A.6}$$

that is, the undesired pseudovector contributions cancel out. For the symmetric traceless part pseudotensor contributions also cancel out, getting eventually

$$S^{XY+YX} = 2I^X S^Y + 2S^X I^Y + (A^X A^Y + (A^X A^Y)^{\text{T}} - \frac{2}{3}\text{Tr}(A^X A^Y)\text{Id}) +$$

$$+ (S^X S^Y + (S^X S^Y)^{\text{T}} - \frac{2}{3}\text{Tr}(S^X S^Y)\text{Id}), \tag{A.7}$$

which has even parity. These results can be summarized as expression (5) in the main text

$$I^{X^{\text{T}}Y^{\text{T}}+Y^{\text{T}}X^{\text{T}}} = I^{XY+YX}, \quad A^{X^{\text{T}}Y^{\text{T}}+Y^{\text{T}}X^{\text{T}}} = -A^{XY+YX}, \quad S^{X^{\text{T}}Y^{\text{T}}+Y^{\text{T}}X^{\text{T}}} = S^{XY+YX}. \tag{A.8}$$

**Invariance of Frobenius norm**

The Frobenius norm of some tensor representation $X$ in TensorNet is invariant under the full orthogonal group O(3). This follows from the cyclic permutation invariance of the trace. As previously shown, parity induces the transposition of $X$, which amounts to a cyclic permutation of the matrix product inside the trace operator, $\mathrm{Tr}(X^\mathrm{T}X) = \mathrm{Tr}(XX^\mathrm{T})$. When considering some rotation $R$, $\mathrm{Tr}(X^\mathrm{T}X) \to \mathrm{Tr}((RXR^\mathrm{T})^\mathrm{T}(RXR^\mathrm{T})) = \mathrm{Tr}(RX^\mathrm{T}R^\mathrm{T}RXR^\mathrm{T}) = \mathrm{Tr}(RX^\mathrm{T}R^{-1}RXR^\mathrm{T}) = \mathrm{Tr}(RX^\mathrm{T}XR^\mathrm{T}) = \mathrm{Tr}(X^\mathrm{T}XR^\mathrm{T}R) = \mathrm{Tr}(X^\mathrm{T}XR^{-1}R) = \mathrm{Tr}(X^\mathrm{T}X)$.

**O(3) to SO(3) symmetry breaking**

The embedding module in TensorNet always produces irreducible representations that are scalars, vectors and tensors, that is, it preserves the parity of the initial components. As previously shown, the computation of the Frobenius norm is O(3)-invariant in this case, making the normalization an operation that also preserves initial parities, since it can be regarded as the modification of the irreducible representations by means of invariant weights. However, depending on the interaction product, O(3)-equivariance can be broken to SO(3)-equivariance in the first interaction layer.

Input irreducible representations for the first interaction layer, which are the outputs of the embedding module, have a well-defined parity. The first normalization that takes place makes use of the Frobenius norm, which at this point is O(3)-invariant, giving O(3)-equivariant representations with the desired parity. Nevertheless, if one considers an interaction product proportional to $Y^{(i)}M^{(i)}$ and computes its irreducible decomposition to generate new features, skew-symmetric and symmetric traceless representations will receive both even and odd parity contributions, as proved above (expressions A.3 and A.4). At this point, one can write some node's full tensor embedding $X$ as

$$X = I + A^+ + A^- + S^+ + S^-, \tag{A.9}$$

where $+$ and $-$ denote even and odd parity, respectively. After obtaining new features through the interaction product, another normalization takes place. The computation of the Frobenius norm $\mathrm{Tr}(X^\mathrm{T}X)$ with current node representations $X$ used to normalize features is no longer O(3)-invariant. One can write

$$\mathrm{Tr}(X^\mathrm{T}X) = \mathrm{Tr}((I + A^+ + A^- + S^+ + S^-)^\mathrm{T}(I + A^+ + A^- + S^+ + S^-)) =$$
$$= \mathrm{Tr}((I - A^+ - A^- + S^+ + S^-)(I + A^+ + A^- + S^+ + S^-)), \tag{A.10}$$

where we have made use of the symmetry or skew-symmetry of the matrices under transposition, and considering the action of parity $A^- \to -A^-, S^- \to -S^-$,

$$\mathrm{Tr}((I - A^+ + A^- + S^+ - S^-)(I + A^+ - A^- + S^+ - S^-)), \tag{A.11}$$

which is manifestly different from A.10 and cannot be reduced to a cyclic permutation of A.10. Thus, the Frobenius norm of representations $X$ at this point of the architecture is not invariant under parity, and therefore normalization amounts to the modification of components with weights $1/(\|X\| + 1)$ which are not invariant under the full O(3), but just under SO(3). From this moment, all features generated in TensorNet will no longer be O(3)-equivariant, but SO(3)-equivariant, and the computation of any Frobenius norm (such as the one used in the scalar output module to predict energies) will only be SO(3)-invariant.

### A.3  Data sets and training details

TensorNet was implemented within the TorchMD-NET framework [41], using PyTorch 1.11.0 [46], PyTorch Geometric 2.0.3 and PyTorch Lightning 1.6.3.

**QM9**

The QM9 [33; 47] data set consists of 130,831 optimized structures of molecules that contain up to 9 heavy elements from C, N, O and F. On top of the structures, several quantum-chemical properties computed at the DFT B3LYP/6-31G(2df,p) level of theory are provided. To be consistent with previous work, we used 110,000 structures for training, which were shuffled after every epoch, 10,000 for validation and the remaining ones were used for testing. We used four random splits, initializing the model with different random seeds. Reported results are the average errors and the standard deviation between parentheses of the last significant digit over these splits.

Models trained with QM9 had 3 interaction layers, 256 hidden channels, used 64 non-trainable radial basis functions and a cutoff of 5Å. The MLP for the embedding part had 2 linear layers, mapping 256 hidden channels as [256, 512, 768], with SiLU activation. The MLPs in the interaction layers had 3 linear layers, mapping the 64 radial basis functions as [64, 256, 512, 768], also with SiLU non-linearities. The scalar output MLP mapped [768, 256, 128, 1] with SiLU activations, giving an atomic contribution to the energy which is added to the atomic reference energies from the data set. All layers were initialized using default PyTorch initialization, except for the last two layers of the output MLP which were initialized using a Xavier uniform distribution and with vanishing biases. Layer normalizations used the default PyTorch parameters. For training, we used a batch size of 16, an initial learning rate of 1e-4 which was reached after a 1000-step linear warm-up, the Adam optimizer with PyTorch default parameters and the MSE loss. The learning rate decayed using an on-plateau scheduler based on the validation MSE loss, with a patience of 15 and a decay factor of 0.8. We did not use an exponential moving average for the validation loss. Inference batch size was 128. Training automatically stopped when the learning rate reached 1e-7 or when the validation MSE loss did not improve for 150 epochs. We used gradient clipping at norm 40. Training was performed on two NVIDIA GeForce RTX 3090 with float32 precision.

### rMD17

The rMD17 [35] data set contains 100,000 recomputed structures of 10 molecules from MD17 [36; 37], a data set of small organic molecules obtained by running molecular dynamics simulations. The DFT PBE/def2-SVP level of theory, a very dense DFT integration grid and a very tight SCF convergence were used for the recomputations. We used 950 and 50 random conformations for training and validation, respectively, and evaluated the error on all remaining conformations. We used five random splits, initializing the model with different random seeds. Reported results are the average errors and the standard deviation between parentheses (when different from zero) of the last significant digit over these splits.

Models trained with rMD17 had 1 and 2 interaction layers, 128 hidden channels, used 32 non-trainable radial basis functions and a cutoff of 4.5Å. The MLP for the embedding part had 2 linear layers, mapping 128 hidden channels as [128, 256, 384], with SiLU activation. The MLPs in the interaction layers had 3 linear layers, mapping the 32 radial basis functions as [32, 128, 256, 384], also with SiLU non-linearities. The scalar output MLP mapped [384, 128, 64, 1] with SiLU activations, giving an atomic contribution to the energy which after addition for all atoms is scaled by the training data standard deviation and shifted with the training data mean. All layers were initialized using default PyTorch initialization, except for the last two layers of the output MLP which were initialized using a Xavier uniform distribution and with vanishing biases. Layer normalizations used the default PyTorch parameters. For training, we used a batch size of 8, an initial learning rate of 1e-3 which was reached after a 500-step linear warm-up, the Adam optimizer with PyTorch default parameters and weighted MSE losses of energy and forces with both weights equal to 0.5. The learning rate decayed using an on-plateau scheduler based on the total weighted validation MSE losses, with a patience of 25 and a decay factor of 0.8. We used an exponential moving average for the energy MSE validation loss with weight 0.99. Inference batch size was 64. Training automatically stopped when the learning rate reached 1e-8 or when the total weighted validation MSE losses did not improve for 300 epochs. We used gradient clipping at norm 40. Training was performed on a single NVIDIA GeForce RTX 2080 Ti with float32 precision.

### SPICE

SPICE [38] is a data set with an emphasis on the simulation of the interaction of drug-like small molecules and proteins. It consists of a collection of dipeptides, drug-like small molecules from PubChem, solvated aminoacids, monomer and dimer structures from DES370K and ion pairs, with a varying number of systems and conformations in each subset, and computed at the $\omega$B97M-D3BJ/def2-TZVPPD level of theory. After filtering molecules containing forces higher than 50.94 eV/Å $\approx$ 1 Hartree/Bohr, SPICE (v 1.1.3) contains approximately 1M data points. We used three random 80%/10%/10% training/validation/test splits, initializing the model with different random seeds. Reported results are the average errors over these splits.

Models trained with SPICE had 2 interaction layers, 128 hidden channels, used 64 non-trainable radial basis functions and a cutoff of 10Å. The MLP for the embedding part had 2 linear layers,

mapping 128 hidden channels as [128, 256, 384], with SiLU activation. The MLPs in the interaction layers had 3 linear layers, mapping the 64 radial basis functions as [64, 128, 256, 384], also with SiLU non-linearities. The scalar output MLP mapped [384, 128, 64, 1] with SiLU activations, giving an atomic contribution to the energy, which after addition for all atoms, the model was trained to match the reference energy of the molecule with subtracted atomic reference energies. All layers were initialized using default PyTorch initialization, except for the last two layers of the output MLP which were initialized using a Xavier uniform distribution and with vanishing biases. Layer normalizations used the default PyTorch parameters. For training, we used a batch size of 16, an initial learning rate of 1e-4 which was reached after a 500-step linear warm-up, the Adam optimizer with PyTorch default parameters and weighted MSE losses of energy and forces with both weights equal to 0.5. The learning rate decayed using an on-plateau scheduler based on the total weighted validation MSE losses, with a patience of 5 and a decay factor of 0.5. We did not use an exponential moving average for the validation loss. Training automatically stopped when the learning rate reached 1e-7 or when the total weighted validation MSE losses did not improve for 50 epochs. Inference batch size was 16. We used gradient clipping at norm 100. Training was performed on 4 NVIDIA GeForce RTX 2080 Ti with float32 precision. The Equivariant Transformer model used for comparison purposes had 4 interaction layers, used 64 non-trainable radial basis functions, 8 attention heads and a cutoff of 10Å.

### ANI1x, COMP6

ANI1x [39; 40] was built using an involved active learning process on the ANI1 data set, giving approximately 5M data points containing organic molecules composed of C, N, O and H. The COMP6 [39] is a benchmarking data set consists of five benchmarks (GDB07to09, GDB10to13, Tripeptides, DrugBank, and ANI-MD) that cover broad regions of organic and biochemical space containing also the elements C, N, O and H, and a sixth one from the previously existing S66x8 noncovalent interaction benchmark. Energies and forces for all non-equilibrium conformations presented were computed at $\omega$B97x59/6-31G(d) level of theory. We trained the model on ANI1x by using three random 80%/10%/10% training/validation/test splits, initializing the model with different random seeds. Reported results on ANI1x and COMP6 are the average errors over these splits.

Models trained with ANI1x had 2 interaction layers, 128 hidden channels, used 32 non-trainable radial basis functions and a cutoff of 4.5Å. The MLP for the embedding part had 2 linear layers, mapping 128 hidden channels as [128, 256, 384], with SiLU activation. The MLPs in the interaction layers had 3 linear layers, mapping the 64 radial basis functions as [64, 128, 256, 384], also with SiLU non-linearities. The scalar output MLP mapped [384, 128, 64, 1] with SiLU activations, giving an atomic contribution to the energy, which after addition for all atoms, the model was trained to match the reference energy of the molecule with subtracted atomic reference energies. All layers were initialized using default PyTorch initialization, except for the last two layers of the output MLP which were initialized using a Xavier uniform distribution and with vanishing biases. Layer normalizations used the default PyTorch parameters. For training, we used a batch size of 64, an initial learning rate of 1e-4 which was reached after a 1000-step linear warm-up, the Adam optimizer with PyTorch default parameters and weighted MSE losses of energy and forces with weights 1 and 100. The learning rate decayed using an on-plateau scheduler based on the total weighted validation MSE losses, with a patience of 4 and a decay factor of 0.5. We did not use an exponential moving average for the validation loss. Training automatically stopped when the learning rate reached 1e-7 or when the total weighted validation MSE losses did not improve for 30 epochs. Inference batch size was 64. We used gradient clipping at norm 100. Training was performed on 4 NVIDIA GeForce RTX 2080 Ti with float32 precision. The Equivariant Transformer [16] model used for comparison purposes had 4 interaction layers, used 64 non-trainable radial basis functions, 8 attention heads and a cutoff of 5Å.

### Scalar, vector and tensor properties of ethanol

We used the reference data for ethanol in vacuum provided together with the FieldSchNet paper [42; 47], in which energies, forces, molecular dipole moments, polarizability tensors and nuclear shielding tensors were computed with the PBE0 functional. To be consistent with results from FieldSchNet [42] and PaiNN [15], 8000 conformations for training and 1000 for both validation and testing were considered. We used three random splits, initializing the model with different random seeds. Reported results are the average errors and the standard deviation between parentheses of the last significant digit over these splits.

As mentioned in the main text, after generating atomic full tensor embeddings $X^{(i)}$, we used different output modules to process their decompositions $I^{(i)}, A^{(i)}, S^{(i)}$ and predict the desired properties. Energies and forces were predicted as described in the scalar output subsection. For the remaining properties:

Molecular dipole moments were predicted by identifying the $n$ skew-symmetric parts $A^{(i)}$ with $n$ vectors $\mathbf{v}^{(i)}$ and mapping with a first linear layer these $n$ vectors to $n/2$ vectors, and then with a second linear layer from $n/2$ vectors to a single vector $\mu^{(i)}$ per atom. In parallel, the norms $||A^{(i)}||$ are fed to a two-layer MLP with SiLU activation to obtain a single scalar per atom $||\mu^{(i)}||$. Then, the molecular dipole moment prediction is obtained by summing over all atoms $\mu = \sum_{(i)} ||\mu^{(i)}||\mu^{(i)}$.

For the prediction of polarizability tensors we used only $I^{(i)}$ and $S^{(i)}$ to directly enforce their symmetry. We fed these components to two sets of two linear layers, to produce two distinct single atomic predictions $\alpha_I^{(i)}$ and $\alpha_S^{(i)}$. Simultaneously, the norms $||I^{(i)} + S^{(i)}||$ were processed by a two-layer MLP with SiLU activation to produce two scalars per atom $||\alpha_I^{(i)}||$ and $||\alpha_S^{(i)}||$. Then, polarizability was predicted by adding all per-atom contributions $\alpha = \sum_{(i)} ||\alpha_I^{(i)}||\alpha_I^{(i)} + ||\alpha_S^{(i)}||\alpha_S^{(i)}$.

Eventually, for the prediction of nuclear shielding tensors, we generated a pseudovector skew-symmetric contribution per atom $A_p^{(i)}$ as described in the Tensor output subsection. We applied three sets of two linear layers to the components $I^{(i)}, A_p^{(i)}, S^{(i)}$ to obtain three single predictions per atom $\sigma_I^{(i)}, \sigma_{A_p}^{(i)}, \sigma_S^{(i)}$. The norms $||I^{(i)} + A_p^{(i)} + S^{(i)}||$ were fed to a two-layer MLP to give three scalars per atom $||\sigma_I^{(i)}||, ||\sigma_{A_p}^{(i)}||, ||\sigma_S^{(i)}||$, used to predict per-atom nuclear shieldings as $\sigma^{(i)} = w^{(i)}(||\sigma_I^{(i)}||\sigma_I^{(i)} + ||\sigma_{A_p}^{(i)}||\sigma_{A_p}^{(i)} + ||\sigma_S^{(i)}||\sigma_S^{(i)})$, where $w^{(i)}$ are element-dependent weights. We used $w^C = 1/0.167, w^O = 1/0.022, w^H = 1$ to account for the different magnitudes of the nuclear shielding tensors, as suggested in [42].

Models trained with ethanol in vacuum had 2 interaction layers, 128 hidden channels, used 32 non-trainable radial basis functions and a cutoff of 4.5Å. The MLP for the embedding part had 2 linear layers, mapping 128 hidden channels as [128, 256, 384], with SiLU activation. The MLPs in the interaction layers had 3 linear layers, mapping the 32 radial basis functions as [32, 128, 256, 384], also with SiLU non-linearities. The scalar output MLP mapped [384, 128, 64, 1] with SiLU activations, giving an atomic contribution to the energy which after addition for all atoms is scaled by the training data standard deviation and shifted with the training data mean, on top of vector and tensor predictions obtained as described above. All layers were initialized using default PyTorch initialization, except for the last two layers of the output MLP which were initialized using a Xavier uniform distribution and with vanishing biases. Layer normalizations used the default PyTorch parameters. For training, we used a batch size of 8, an initial learning rate of 1e-3 without warm-up, the Adam optimizer with PyTorch default parameters and weighted MSE losses of energy, forces, dipole moments, polarizabilities and nuclear shieldings, with weights of $0.5\times(627.5)^2$, $0.5\times(1185.82117)^2$ (the squared conversion factors from Ha to kcal/mol and Ha/Bohr to kcal/mol/Å), 100, 100 and 100, respectively. The learning rate decayed using an on-plateau scheduler based on the total weighted validation MSE losses, with a patience of 30 and a decay factor of 0.75. We used an exponential moving average for the energy MSE validation loss with weight 0.99. Training automatically stopped when the learning rate reached 1e-8 or when the total weighted validation MSE losses did not improve for 300 epochs. Inference batch size was 64. We used gradient clipping at norm 100. Training was performed used the original data atomic units, performing unit conversion at test stage, using two NVIDIA GeForce RTX 2080 Ti with float32 precision. Errors in chemical shifts were computed as MAEs of mean traces of predicted and target nuclear shielding tensors.

**Equivariance, interaction and cutoff ablations**

We used the rMD17 data set and training details for both aspirin and toluene, varying only the number of layers, the cutoff radius and the interaction products, as detailed in the experiment. We used three random splits, initializing the model with different random seeds. Reported results are the average errors over these splits.

**Computational cost**

TorchScript code optimization and PyTorch (2.0) compilation were not used for the experiments. For TensorNet, we used the model with rMD17 architectural specifications. The ET used 128 hidden channels, 32 radial basis functions and 8 attention heads. For inference time, we used the blocked autorange benchmarking functionality in PyTorch with a minimum runtime of 10. For training step and epoch timings, we waited until the value stabilized. All experiments were performed with float32 precision.

