# OpenReview forum: "TensorNet: Cartesian Tensor Representations for Efficient Learning of Molecular Potentials"
_NeurIPS.cc/2023/Conference — NeurIPS 2023 poster_

### Official Review · Reviewer_QwGa · 2023-07-03

**Soundness:** 3 good
**Presentation:** 3 good
**Contribution:** 3 good
**Rating:** 6
**Confidence:** 2

**Summary:**

The paper presents TensorNet, a novel O(3)-equivariant message-passing neural network for efficient representation of molecular systems in scientific research. This model utilizes Cartesian tensor atomic embeddings to simplify feature mixing via matrix product operations. By decomposing tensors into rotation group irreducible representations, it enables independent processing of scalars, vectors, and tensors when required. TensorNet outperforms higher-rank spherical tensor models in performance, utilizing fewer parameters, even with a single interaction layer for small molecule potential energies. Additionally, TensorNet can accurately predict vector and tensor molecular quantities on top of potential energies and forces, greatly reducing the model's computational cost. Therefore, TensorNet provides a promising framework for developing state-of-the-art equivariant models with enhanced efficiency and computational affordability.

**Strengths:**

1. This paper adeptly presents TensorNet, a new learning model that not only establishes state-of-the-art performance but does so with a remarkable reduction in the number of parameters utilized. This marks a significant leap in model efficiency without sacrificing performance quality, setting a new benchmark in the field.

2. The majority of the empirical outcomes display a marked enhancement over existing methods, with the implementation of TensorNet consistently yielding superior results. The experimental evidence provided substantiates the model's efficacy, reinforcing the robustness and applicability of this innovative approach in real-world scenarios.

**Weaknesses:**

The paper's exposition of the model architecture can be somewhat challenging to comprehend due to its complex nature. A potential improvement would be the inclusion of intuitive diagrams or visual aids within the main body of the text, not just in the Appendix. Simplified illustrations, possibly even a step-by-step visual guide, could greatly enhance the reader's understanding of the architecture and make the methodology more accessible to a broader audience.

**Questions:**

1. The proposed architecture (illustrated in Fig. A1) appears quite intricate. Could you provide more insight or intuitive reasoning behind the specific design choices in the model? What fundamental principles or considerations have influenced the complexity and uniqueness of this architecture?

2. Could you delve deeper into the primary constraints or drawbacks of the TensorNet model compared to the existing methods in the field? What are the potential areas where other models might still hold an advantage, and how does TensorNet aim to address these challenges in future iterations or improvements?

---

> ### Author Rebuttal · Authors · 2023-08-08
>
> We thank the reviewer for their positive feedback on the model. We now address the questions raised.
>
> **The proposed architecture (illustrated in Fig. A1) appears quite intricate. Could you provide more insight or intuitive reasoning behind the specific design choices in the model? What fundamental principles or considerations have influenced the complexity and uniqueness of this architecture?**
>
> We will address this issue in terms of clarity of the figure and the model by adding an additional figure (see general rebuttal’s pdf).
>
> One can justify the design choices by taking into account the precise operations that allow TensorNet to be O(3) equivariant and taking them as the fundamental building blocks:
>
> 1) Linear combinations of irreducible components: mixing independently different rank features (different linear layers for I, A and S) is presumed to give more flexibility to the model when compared to just applying a linear layer to X (= I+A+S).
>
> 2) Multiplication with invariant quantities: again, we assume it is preferable to modify independently I, A, S compared to modifying X alone. We can distinguish subcases:
>
> - In the case of the embedding module: We need to generate scalars, vectors and tensors from relative position vectors, that is, edge-wise. Assuming more model flexibility when modifying independently I, A, S (compared to just modifying X) by multiplying invariant quantities, it makes sense to encode edge interatomic distance with different learnable functions $f^{(ij)}_I$, $f^{(ij)}_A$, $f^{(ij)}_S$. After aggregation of edge-wise features into node-wise features, we use again the modification with invariants by taking the (normalized) norm and obtaining new invariants through a MLP that will modify the node’s I, A, S, enabling further processing of the direct neighborhood that has been aggregated.
>
> - In interaction layers, when considering the message-passing framework, we find that encoding interatomic distance (weighted with a cutoff) separately to incoming I, A, S features from neighbors goes in line with previous message-passing models for molecular potentials while exploiting the flexibility of independent modifications of different rank features.
>
> 3) Interaction via matrix product: We prove that using a particular sum of products the model exhibits O(3) equivariance. Importantly, these products are performed at the node level, after aggregation, which is more efficient. There are an infinite amount of expressions in terms of matrix products that allow O(3) equivariance: for example, w * YM + w * MY, where w can be a learnable vector of weights with hidden channels length and * is the element-wise product. The most simple one, 1 * YM + 1 * MY, is the one currently used.
>
> 4) We can consider separately the normalization operations that take place. There are two normalizations:
>
> - Using LayerNorm: It can be expected that when taking the norms of I, A and S, their magnitude can be quite different. LayerNorm would equilibrate the norms coming from I, A and S. In fact, the LayerNorm used in the output MLP giving energies gave a substantial benefit in terms of accuracy.
>
> - Normalizing X tensors by means of X -> X / (||X||+1): we found this to stabilize training (without this normalization, for some molecules the loss became NaN), +1 ensures numerical stability (though any other number could be used). This normalization also enforces a non-linearity. Since ||X|| = || I + A + S || is used (as opposed to using separately || I ||, ||A||, ||S||) the components I, A, S receive non-linear contributions from the other components.
>
> We hope this explanation helps in clarifying the architectural choices, given that the use of the different components I, A, S along the architecture makes the diagram convoluted.
>
> **Could you delve deeper into the primary constraints or drawbacks of the TensorNet model compared to the existing methods in the field? What are the potential areas where other models might still hold an advantage, and how does TensorNet aim to address these challenges in future iterations or improvements?**
>
> The main limitation comes from the prediction of quantities built on top of higher-rank tensors, like functions expanded in terms of spherical harmonics, e.g. electronic densities. However, TensorNet can still predict the density as a scalar on each point, the limitation comes from the representation in terms of higher-rank tensors (bearing in mind that our ultimate intention is to predict energies and forces in a fast and accurate way, and therefore this is not a requirement). The prediction of these quantities of rank higher than two is very uncommon. Examples of physical quantities that can correctly be predicted up to rank-2 apart from energies and forces are molecular or atomic dipoles, polarizability tensors, nuclear-shielding tensors, and quadrupole moments, which account for the vast majority of quantities that are used in molecular settings.
>
> We do think that TensorNet will be primarily used to predict neural network potentials (molecular energies and forces), and molecular properties which is the reason for which it has been designed and where it holds its main advantages. e3nn based models will still be used where speed is less important or somehow the higher rank tensors are required, e.g. for representing functions in terms of spherical harmonics.
>
> For the future, we are currently exploiting the simple matrix operations of TensorNet further to achieve even greater performance speed-ups, mainly using Cuda graphs and other standard techniques. Integration with a widely used biomolecular dynamics package is already in progress with the final aim to be used in practical applications.

---

> > ### Comment · Reviewer_QwGa · 2023-08-16
> >
> > I thank the authors for responding to my concern and for promising the improvement of the paper.

---

### Official Review · Reviewer_u3os · 2023-07-07

**Soundness:** 3 good
**Presentation:** 2 fair
**Contribution:** 3 good
**Rating:** 6
**Confidence:** 2

**Summary:**

This paper proposes TensorNet, an O(3)-equivariant neural network architecture for molecules. Using the decomposition of a 3x3 matrix into a scalar, vector, and matrix shown in Eq. (2), TensorNet efficiently computes the interaction of O(3)-equivariant features up to l=2, where l is the degree (frequency) of the O(3) representation. The performance is evaluated on several standard benchmarks, such as qm9, which show that TensorNet achieves equivalent or better performance than baselines.

**Strengths:**

1. TensorNet is a novel method. I have never seen the construction of an equivariant net based on the decomposition (2).
2. The performance is evaluated with several different datasets and in terms of different metrics (e.g., prediction error, computation speed).
3. The performance is comparable to or better than existing approaches.

**Weaknesses:**

1. The paper has room to improve in terms of presentation. I'm unfamiliar with the chemistry (molecule) domain, and some parts seem challenging to understand without expertise. For example, a vector r_ij is defined on line 175, but the mathematical definition is not described. Also, the meaning of the cutoff radius is unexplained. Another point is that there is no reference nor citation to Eq. (2), the core equation of this paper. These are not well known in the machine learning community, and it could be better to introduce them in plain words.
2. The limitations of the proposed method are not explicitly discussed. One limitation is that TensorNet cannot capture higher degree (l>2) information of O(3).
3. It is argued that some existing methods have a downside: "the computation of tensor products in most of these models containing higher-rank tensors and pseudotensors can be expensive" (lines 111-112). However, only one method (ET) is compared with the proposed method in terms of computational cost. Also, there is no theoretical evaluation of the complexity e.g., using big-O notation.

**Questions:**

n/a

**Limitations:**

Limitations are not explicitly addressed. No particular concern for potential negative societal impact.

---

> ### Author Rebuttal · Authors · 2023-08-08
>
> We thank the reviewer for their positive feedback on the novelty of the approach, completeness of the validation, and results. We here answer the further points raised.
>
> **The paper has room to improve in terms of presentation. I'm unfamiliar with the chemistry (molecule) domain, and some parts seem challenging to understand without expertise. For example, a vector r_ij is defined on line 175, but the mathematical definition is not described. Also, the meaning of the cutoff radius is unexplained. Another point is that there is no reference nor citation to Eq. (2), the core equation of this paper. These are not well known in the machine learning community, and it could be better to introduce them in plain words.**
>
> We understand the concern that the reviewer mentions about some definitions not being explicitly addressed for researchers not familiar with the molecular domain. We will address this and improve it in the new version of the manuscript. It will also be clarified in the additional figure that would be incorporated into the manuscript (see general rebuttal’s pdf).
>
> The edges between nodes (atoms) are built by defining a cutoff radius, where atoms $i$ and $j$ are connected by an edge $ij$ if they are at a distance smaller than the cutoff radius. At the same time, for every edge, we can consider a relative position vector $r_{ij}$, computed from 3D coordinates of atoms $r_i$ and $r_j$, as $r_{ij}$ = $r_j$ - $r_i$.
>
> Regarding the decomposition (2), the reference is [21] (page 3, first and second paragraphs), even though it is true that it is found two or three lines below the expression and it may not be immediately clear that it refers to (2). Therefore, we will add an additional reference and citation to clarify.
>
> **The limitations of the proposed method are not explicitly discussed. One limitation is that TensorNet cannot capture higher degree (l>2) information of O(3).**
>
> Regarding limitations, we will include them as an independent section at the end of the manuscript.
>
> TensorNet is not using tensors of rank higher than 2 to favor computational efficiency. In terms of accuracy for lower-rank quantities, this does not seem to be a problem, as results show that TensorNet performs competitively or better compared to models using higher ranks. TensorNet cannot choose the maximum rank of the tensors being used,  in contrast to spherical models, where the rank can be seen as a hyperparameter, and the architectures are built in a general way that can consistently include higher ranks. The main limitation comes from the prediction of quantities built on top of higher-rank tensors, like functions expanded in terms of spherical harmonics, e.g. electronic densities. However, TensorNet can still predict the density as a scalar on each point, the limitation comes from the representation in terms of higher-rank tensors (bearing in mind that our ultimate intention is to predict energies and forces in a fast and accurate way, and therefore this is not a requirement). The prediction of these quantities of rank higher than two is uncommon. Examples of physical quantities that can correctly be predicted by TensorNet up to rank-2 are energies, forces, molecular or atomic dipoles, polarizability tensors, nuclear-shielding tensors, and quadrupole moments, which account for the majority of quantities that are used in molecular systems.
>
> **It is argued that some existing methods have a downside: "the computation of tensor products in most of these models containing higher-rank tensors and pseudotensors can be expensive" (lines 111-112). However, only one method (ET) is compared with the proposed method in terms of computational cost. Also, there is no theoretical evaluation of the complexity e.g., using big-O notation.**
>
> The computational cost of models based on spherical tensors is well-established and we will add more comparisons in the manuscript. For example, in reference [20] (MACE), a fast model is built that explicitly addresses this issue, and one can see in the reference in Table 2, at the end of page 8, times to compute energy and forces are compared for NequIP [17], BOTNet [19] and MACE on the 3BPA molecule with 27 atoms. MACE is 4x faster than previous models when considering the full model (L=2, the SOTA model for rMD17). TensorNet 2L (our SOTA model for rMD17) with a batch size of 32 is ~4x faster than MACE on the same molecule and batch size, on a NVIDIA RTX 2080 Ti, and therefore it is by far faster than NequIP and BOTNet.
>
> We compared our speed to the ET [16], a Cartesian vector equivariant model, as opposed to the other higher-rank spherical models, because Cartesian vector representations are found to be fast (also, notice that the SOTA ET model had 6 layers, and we compare to 4 and 5 layers). Our intention is to emphasize that TensorNet, with accuracies comparable to SOTA spherical models, is as computationally efficient (or even more computationally efficient in some cases) than a Cartesian vector model for small molecules.
>
> Finally, the complexity of TensorNet is linear regardless of the rank with the number of atoms N (~O(N)), which is the main scaling parameter. The number of neighbors per atom M is controlled by the cutoff radius and the density, which are usually fixed across models and systems, therefore it is not usually a scaling factor to take into account, but rather a constant factor that is included in the overall speed. In any case, since matrix products are performed after aggregation over neighbors (we compute matrix products node-wise, not edge-wise), these do not scale with M. This is in contrast to spherical models, where tensor products are computed on edges, and therefore display a worse scaling with the number of neighbors M, that is, a worse scaling when increasing the cutoff radius at fixed density. We will add this to the manuscript.

---

> > ### Comment · Reviewer_u3os · 2023-08-12
> > **Response**
> >
> > Thank you for the response. The rebuttal comments make sense to me (especially the reason of rank 2) and mostly resolve my concerns. I will raise my score.

---

### Official Review · Reviewer_3Tca · 2023-07-07

**Soundness:** 3 good
**Presentation:** 3 good
**Contribution:** 3 good
**Rating:** 5
**Confidence:** 4

**Summary:**

This paper proposes a cartesian tensor representation for efficient learning of molecular potentials. It enables the feature mixing process to be a simple matrix product operation. In addition, the matrix product operation is simplified by cost-effective decomposition techniques. Experimental results demonstrate that the proposed method can effectively reach a comparable performance with a much smaller number of parameters.

**Strengths:**

(1) The proposed method is very technically solid and this reviewer also thinks the efficient computation of equivariant architectures becomes increasingly important;

(2) The extension of torchMD-net with efficient decomposition techniques is reasonably motivated, just like low-rank decomposition techniques for large language models;

(3) Experimental settings are very solid, covering enough number of experimental settings, which fully support the effectiveness of the proposed method.

**Weaknesses:**

(1) The presentation is not that easy to follow and the notation is a little bit complicated, which brings additional hardness for readers to understand the core idea. In addition, although the paper is more about mathematical techniques utilization, no figure illustration provided is still very tough for readers to quickly capture the core idea;

(2) The proposed method is more like an application of efficient tensor decomposition techniques to the existing equivariant network architectures. Without molecular domain-specific insights somehow lowers the significance of the proposed method.

**Questions:**

It seems the proposed architecture is an extension on TorchMD-net. Is it possible to apply this refinement to other equivariant models? If not, then this limitation will significantly lower its significance.

**Limitations:**

NA.

---

> ### Author Rebuttal · Authors · 2023-08-08
>
> We thank the reviewer for the nice feedback on the strengths of TensorNet. We clarify here the other points raised.
>
> **The presentation is not that easy to follow and the notation is a little bit complicated, which brings additional hardness for readers to understand the core idea. In addition, although the paper is more about mathematical techniques utilization, no figure illustration provided is still very tough for readers to quickly capture the core idea**
>
> While the mathematics can be a bit overwhelming, we tried to make it as simple as possible using the supplementary information for the more technical parts. We do provide a figure illustration in supplementary, however, we will introduce another more illustrative figure (see general rebuttal’s pdf) and, space permitting, include the one found in supplementary in the main manuscript. Also, we will use the extra space to further clarify the explanation of the mathematical techniques.
>
> **The proposed method is more like an application of efficient tensor decomposition techniques to the existing equivariant network architectures. Without molecular domain-specific insights somehow lowers the significance of the proposed method**
>
> To our understanding, TensorNet, apart from being a mathematical framework, contains physical inductive biases which are meaningful in molecular domain-specific settings. A common approach in molecular physics is to expand some interaction energy in terms of charges (scalars), dipoles (vectors), quadrupoles (rank-2 tensors) and so on, see for example reference [13] (sections 2 and 3). In TensorNet, atomic features are learnable full tensors that depend on neighboring atoms and that can be decomposed precisely into scalars, vectors, and tensors which interact with other atoms’ features by means of matrix products, giving rise to new scalar, vector and tensor features. These products can be regarded as computing all possible combinations (interactions) between scalars, vectors and tensors, akin to scalar/dipole, dipole/dipole, dipole/quadrupole and quadrupole/quadrupole interactions (and so on) in a unified way, contributing to the predicted potential energy of the system.
>
> In fact, we also show that for ethanol in a vacuum, TensorNet can simultaneously and accurately predict physical quantities of different geometrical nature from shared atomic features, pointing to the fact that these features are physically meaningful. These facts will be reinforced in the manuscript.
>
> **It seems the proposed architecture is an extension on TorchMD-net. Is it possible to apply this refinement to other equivariant models? If not, then this limitation will significantly lower its significance.**
>
> TensorNet is not an extension of the Equivariant Transformer, but a completely new model built on the same codebase. TorchMD-NET is a highly optimized PyTorch-based library for neural network potentials, in which the Equivariant Transformer (ET) [16] (probably the model the reviewer refers to when mentioning TorchMD-NET) is one of several existing models. Other models include a graph neural network similar to SchNet, an invariant transformer, and currently TensorNet.
>
> In fact, any equivariant model based on Cartesian vectors (such as PaiNN [15] and the ET [16]), can be rewritten using TensorNet’s formalism, by identifying scalar features with tensor features (matrix features) proportional to the identity matrix, and vector features with skew-symmetric tensors. As the reviewer mentions, these models lack the ‘refinement’ of the incorporation of rank-2 features. Given the current implementation of Cartesian vector equivariant models, separated into scalar and vector pathways, one could consider the incorporation of a rank-2 feature pathway. In this regard, TensorNet provides a framework that allows working in a unified way with ‘general’ geometrical objects (full tensors) and their decomposition into scalars, vectors, and tensors, as opposed to designing differentiated pathways for different-rank features.

---

### Official Review · Reviewer_1mQm · 2023-07-11

**Soundness:** 3 good
**Presentation:** 3 good
**Contribution:** 1 poor
**Rating:** 2
**Confidence:** 4

**Summary:**

The paper introduces Tensor3 Net, a message-passing neural network architecture designed for molecular systems representation. Tensor3 Net leverages rank-2 Cartesian tensor representations and O(3)-equivariance. The tensors are decomposed into rotation group irreducible representations, enabling separate processing of scalars, vectors, and tensors when necessary.

**Strengths:**

The paper is well-written and the idea of building a O(3) equivariant model with Cartesian Tensor is valid, given the computation complexity of models such as e3nn, based on spherical harmonics.

**Weaknesses:**

The one main weakness is that the author only model rank-2 tensors, which is relatively arbitrary. What it should be done is to introduce a network with general rank cartesian tensors (these will then be 3 x 3 x 3 x ... x 3 tensors). I see the value of adding the l=2 components in a cartesian way too marginal for publication at Neurips.

**Questions:**

See weaknesses.

**Limitations:**

The authors do not discuss the limitations of their approach.

---

> ### Author Rebuttal · Authors · 2023-08-08
>
> We thank the reviewer for the feedback on the model in terms of its validity compared to e3nn. Here we address the main point raised.
>
> **The one main weakness is that the author only model rank-2 tensors, which is relatively arbitrary. What it should be done is to introduce a network with general rank cartesian tensors (these will then be 3 x 3 x 3 x ... x 3 tensors). I see the value of adding the l=2 components in a cartesian way too marginal for publication at Neurips.**
>
> TensorNet proposes a novel idea on how to achieve O(3)-equivariance using simple and fast matrix operations. These operations are commonly executed in neural networks and fast on GPUs.
>
> Even though we are limiting the operations to rank-2 Cartesian tensors, the model achieves state-of-the-art accuracies or performs better than models based on e3nn or using up to rank-3 spherical tensors (and pseudotensors). The use of rank-3 spherical tensors renders those models more computationally expensive, and in fact, no neural network potentials have used a rank higher than 3, while they have the possibility of using arbitrary higher ranks (of course, at extra cost). TensorNet achieves the same accuracy as models using higher-rank tensors (and pseudotensors).
>
> The results are important because if we want to enable molecular dynamics simulations using neural network potentials at quantum-level of accuracy, we need to compute forces for millions and even billions of time steps and fast potentials are critical for that.  We foresee TensorNet’s approach to be important in the quest to replace molecular mechanics potentials commonly used in bio-molecular dynamics.
> In fact, any equivariant model based on Cartesian vectors (such as PaiNN [15] and the ET [16]), can benefit from the idea and be rewritten using TensorNet’s formalism, by identifying scalar features with tensor features (matrix features) proportional to the identity matrix, and vector features with skew-symmetric tensors. Given the current implementation of Cartesian vector equivariant models, separated into scalar and vector pathways, one could consider the incorporation of a rank-2 feature pathway. In this regard, TensorNet provides a framework that allows working in a unified way with ‘general’ geometrical objects (full tensors) and their decomposition into scalars, vectors, and tensors, as opposed to designing differentiated pathways for different-rank features.
>
> The decomposition of arbitrary rank Cartesian tensors into irreducible representations of the rotation group is highly non-trivial. For the case 3x3x3 (rank 3), it can be shown (see for example the paper *‘Decomposition of third-order constitutive tensors’* from Y. Itin and S. Reches, available on arXiv, subsection 4.2, and Figure 1 on page 24) that the 3x3x3 = 27-dimensional representation can be decomposed into 1 + 3 + 3 + 3 + 5 + 5 + 7, that is a scalar, three vectors, two quadrupoles, and an octupole (as opposed to rank-2 decomposition, 3x3 = 9 = 1 + 3 + 5). The decomposition is very complex (subsection 4.2) and involves a significant amount of operations. These decompositions for arbitrary ranks would be unfeasible in terms of memory and computational efficiency. In contrast, for rank-2 Cartesian tensors, the decomposition only requires computing a trace and a matrix transpose and already achieves SOTA.
>
> In terms of limitations, constraining TensorNet to rank-2 restricts the prediction of quantities built on top of higher-rank tensors, like functions expanded in terms of spherical harmonics, e.g. electronic densities. However, TensorNet can still predict the density as a scalar on each point, the limitation comes from the expansion in terms of higher-rank tensors. We want to emphasize that our ultimate intention is to predict energies and forces in a fast and accurate way, and therefore this would not strictly be a requirement. Also, the prediction of these quantities of rank higher than two is uncommon. Examples of physical quantities that can correctly be predicted up to rank-2 apart from energies and forces are molecular or atomic dipoles, polarizability tensors, nuclear-shielding tensors, and quadrupole moments, which account for the vast majority of quantities that are used in molecular settings.
>
> In summary, we believe that TensorNet deserves publication here because it is a novel idea that will be very useful to be integrated by any models aiming for O(3) equivariance using fast operations. The rank-2 restriction does not limit the applicability of the model due to SOTA accuracy and the wide range of physical quantities to which it can be applied.

---

### Author Rebuttal · Authors · 2023-08-09

The questions raised by the reviewers have been addressed in their individual rebuttals. However, since some of them expressed concerns about the clarity of exposition, we use this general rebuttal to attach a figure that will be added to the manuscript, which we hope provides clarification on both the notation and the methods.

---

### Decision · Program_Chairs · 2023-09-21

**Decision:**

Accept (poster)

**Comment:**

The paper proposes a neural network architecture that is invariant to SO(3) transformations. The key idea is to represent matrices in a rotation invariant name (named 'rank-2 tensors', but for ML community such notation is not very standard). The reviewers agree that paper is well-written and the results are quite convincing. The main concern (I also share it) is that the paper only works with matrices, not going into higher-dimensional tensors. The authors, however, have provided a rebuttal where they explain the theory behind that. Unfortunately, the reviewer have not engaged back into the discussion, thus I base my decision on my own reading and understanding of the paper, as well as other reviews, which are positive.